# Spearheading future omics analyses using dyngen, a multi-modal simulator of single cells

Robrecht Cannoodt [1,2,3,5], Wouter Saelens[1,2,4,5], Louise Deconinck [1,2] & Yvan Saeys [1,2 ✉]

We present dyngen, a multi-modal simulation engine for studying dynamic cellular processes at single-cell resolution. dyngen is more flexible than current single-cell simulation engines, and allows better method development and benchmarking, thereby stimulating development and testing of computational methods. We demonstrate its potential for spearheading computational methods on three applications: aligning cell developmental trajectories, cell-specific regulatory network inference and estimation of RNA velocity.

---

[1] Data Mining and Modelling for Biomedicine group, VIB Center for Inflammation Research, Ghent, Belgium. [2] Department of Applied Mathematics, Computer Science, and Statistics, Ghent University, Ghent, Belgium. [3] Data Intuitive, Lebbeke, Belgium. [4] Institute of Bioengineering, School of Life Sciences, École Polytechnique Fédérale de Lausanne (EPFL), Lausanne, Switzerland. [5]These authors contributed equally: Robrecht Cannoodt, Wouter Saelens. ✉email: yvan.saeys@ugent.be

Single-cell simulation engines are becoming increasingly important for testing and benchmarking computational methods, a pressing need in the widely expanding field of single-cell biology. Complementary to real biological data, synthetic data provides a valuable alternative where the actual ground truth is completely known and thus can be compared to, in order to make quantitative evaluations of computational methods that aim to reconstruct this ground truth[1]. In addition, simulation engines are more flexible when it comes to stress-testing computational methods, for example by varying the parameters of the simulation, such as the amount of noise, samples, and cells measured, allowing benchmarking of methods over a wide range of possible scenarios. In this way, they can even guide the design of real biological experiments, finding out the best conditions to be used as input for subsequent computational pipelines.

Another, more experimental use of simulation engines is their important role in spearheading the development of computational methods, possibly even before real data is available. In this way, simulation engines can be used to assess the value of novel experimental protocols or treatments. Simulation engines are also increasingly important when it comes to finding alternatives to animal models, for example for drug testing and precision medicine. In such scenarios, cellular simulations can act as digital twins, offering unlimited experimentation in silico[2].

Simulating realistic data requires that the underlying biology is recapitulated as best as possible, and in the case of transcriptomics data this typically involves modelling the underlying gene regulatory networks. Simulators of "bulk" microarray or RNA-sequencing profiles simulate biological processes (e.g. transcription, translation) by translating a database of known regulatory interactions into a set of ordinary differential equations (ODE)[3–6]. These methods have been instrumental in performing benchmarking studies[7–9]. However, the advent of single-cell omics introduced several new types of analyses (e.g. trajectory inference, RNA velocity, cell-specific network inference) which exploit the higher resolution of single-cell versus bulk omics[10]. In addition, the data characteristics of single-cell omics are vastly different from bulk omics, typically having much lower library sizes and a higher dropout rate, but also a high number of profiles[11]. The low library sizes, in particular, are problematic as ODEs are ill-suited for performing low-molecule simulations[12]. This necessitates the development of single-cell simulators.

To this end, single-cell omics simulators emulate the technical procedures from single-cell omics protocols. Simulators such as Splatter[1], powsimR[13], PROSSTT[14], and SymSim[15]) have already been widely used to compare single-cell methods[16–19] and perform independent benchmarks[20–22]. However, by focusing more on simulating the single-cell omics protocol (e.g. RNA capture, amplification, sequencing) and less on the underlying biology (e.g. transcription, splicing, translation), their applicability and reusability is limited towards the specific application for which they were designed (e.g. benchmarking clustering or differential expression methods), and extending these tools to include additional modalities or experimental conditions is challenging.

We introduce dyngen, a method for simulating cellular dynamics at a single-cell, single-transcript resolution (Fig. 1). This problem is tackled in three fully configurable main steps. First, biological processes are mimicked by translating a gene regulatory network into a set of reactions (regulation, transcription, splicing, translation). Second, individual cells are simulated using Gillespie's stochastic simulation algorithm (SSA)[12], which is designed to work well in low-molecule simulations. Finally, real reference datasets are used to emulate single-cell omics profiling protocols. Throughout a simulation, dyngen tracks many layers of information, including the abundance of any molecule in the cell,

the progression of the cell along a dynamic process, and the activation strength of individual regulatory interactions. In addition, dyngen can simulate a large variety of dynamic processes (e.g. cyclic, branching, disconnected) as well as a broad range of experimental conditions (e.g. batch effects and time-series, perturbation and single-cell knockdown experiments). For these reasons, dyngen can cater to a wide range of benchmarking applications, including trajectory inference, trajectory alignment, and trajectory differential expression (Supplementary Table 1).

## Results
We demonstrate dyngen's broad applicability by evaluating three types of computational approaches for which no simulation engines exist yet: cell-specific network inference, trajectory alignment and RNA velocity (Fig. 2). We emphasise that our main aim here is to illustrate the potential of dyngen for these evaluations, rather than performing large-scale benchmarking, which would require assessing many more quantitative and qualitative aspects of each method[23].

Use-case "trajectory alignment". Trajectory alignment methods align trajectories from different samples and allow studying the differences between the different trajectories. For example, by comparing the transcriptomic profiles of cells from a diseased patient to a healthy control, it might be possible to detect transcriptomics differences (differential expression) of particular cells along a developmental process, or to detect an early stop of the trajectory of the diseased patient. Currently, trajectory alignment is limited to aligning linear trajectories, though other topologies of a trajectory could be aligned as well. Dynamic Time Warping (DTW)[24] is a method designed for aligning temporal sequences for speech recognition but has since been used to compare gene expression kinetics from many different biological processes[25–28]. cellAlign[28] uses DTW to perform trajectory alignment, but also includes interpolation and scaling of the single-cell data as a preprocessing step. We evaluate the performance of DTW and cellAlign by simulating 40 datasets, each containing two linear trajectories generated with the same gene regulatory network but with slightly different simulation kinetics. We assess the accuracy of the obtained alignments by comparing the generated alignment path with the worst possible alignment that could be performed (Supplementary Fig. 1D), named the Area Between Worst And Prediction (ABWAP) score. Overall, cellAlign performs significantly better than DTW (Supplementary Fig. 1), which is likely due to the interpolation and scaling steps provided by cellAlign, reducing noise in the data and improving the comparability of the trajectories. Note that, in this comparison, only linear trajectory alignment is performed. While dyngen can generate non-linear trajectories (e.g. cyclic or branching), both aligning non-linear trajectories and constructing a quantitative accuracy metric for non-linear trajectory alignment is not trivial and an avenue for future work.

Use-case "RNA velocity". RNA velocity methods use the relative ratio between pre-mRNA and mature mRNA reads to predict the rate of increase/decrease of RNA molecule abundance, as this can be used to predict the directionality of single-cell differentiation in trajectories[29,30]. Already two algorithms are currently available for estimating the RNA velocity vector from spliced and unspliced counts: velocyto[30] and scvelo[31]. Yet, to date, no quantitative assessment of their accuracy has been performed, mainly due to the difficulty in obtaining real ground-truth data to do so. In contrast, the ground-truth RNA velocity can be easily extracted from a dyngen simulation, as it is possible to store the rate at which mRNA molecules are being transcribed and degraded at any particular point in time. We executed velocyto and scvelo (with 2 different parameter settings, stochastic and

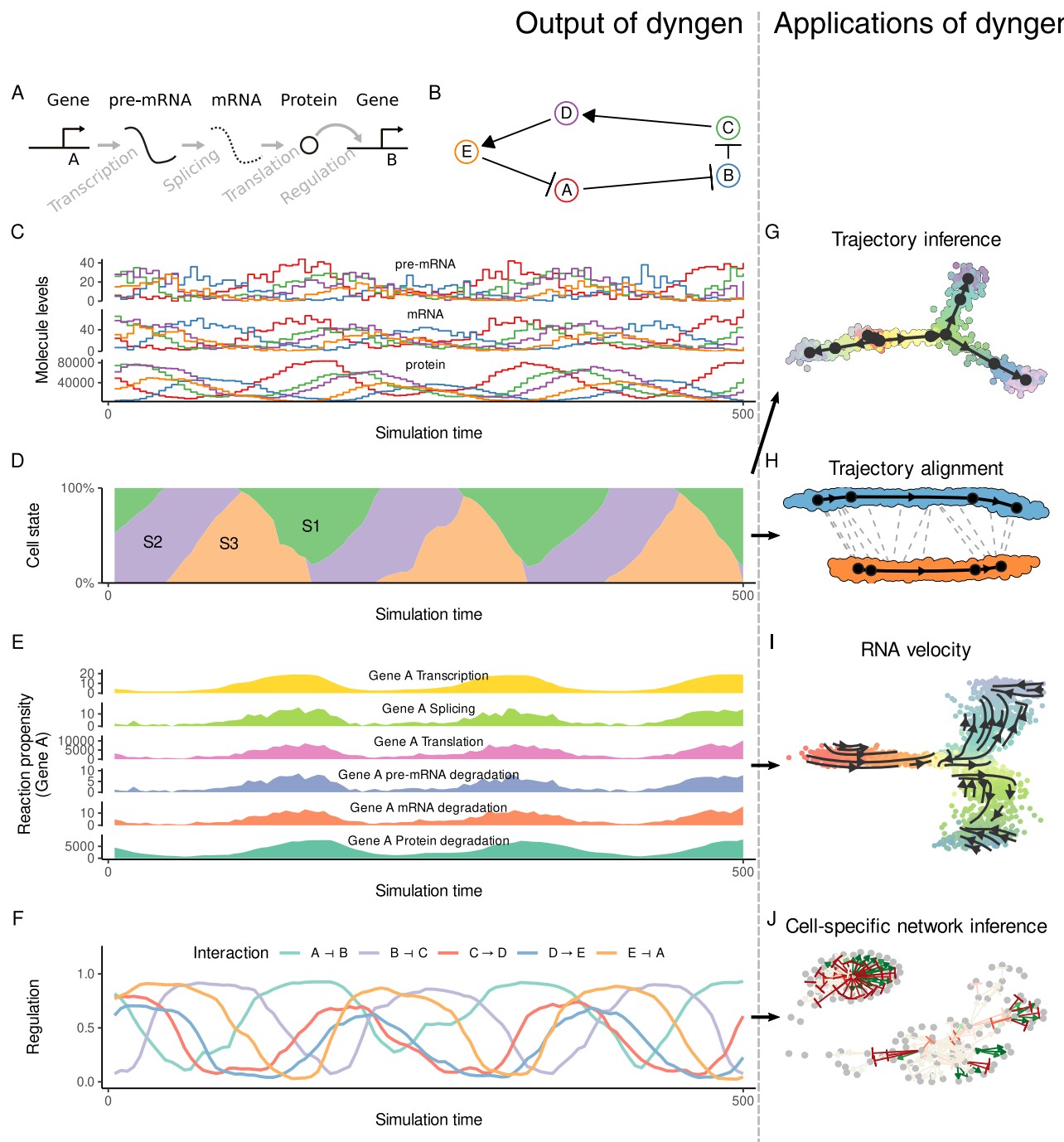

**Fig. 1 Showcase of dyngen functionality. A** Changes in abundance levels are driven strictly by gene regulatory reactions. **B** The input Gene Regulatory Network (GRN) is defined such that it models a dynamic process of interest. **C** The reactions define how abundance levels of molecules change at any particular time point. **D** Firing many reactions can significantly alter the cellular state over time. **E** dyngen keeps track of the likelihood of a reaction firing during small intervals of time, called the propensity, as well as the actual number of firings. **F** Similarly, dyngen can also keep track of the regulatory activity of every interaction. **G** A benchmark of trajectory inference methods has already been performed using the cell state ground-truth[21]. **H** The cell state ground-truth enables evaluating trajectory alignment methods. **I** The reaction propensity ground-truth enables evaluating RNA velocity methods. **J** The cellwise regulatory network ground-truth enables evaluating cell-specific gene regulatory network inference methods.

dynamical) on 42 datasets with a variety of backbones (including linear, bifurcating, cyclic, disconnected). We evaluated the predictions using two metrics (Supplementary Fig. 2), one which directly compares the predicted RNA velocity of each gene with the ground-truth RNA velocity (called the "velocity correlation"), and one which compares the direction of the ground-truth trajectory embedded in a dimensionality reduction with the average RNA velocity of cells in that neighbourhood (called the "velocity

arrow cosine"). While both velocyto and scvelo obtained high scores for the velocity arrow cosine metric (overall 25th percentile = 0.606), the velocity correlation is rather low (overall 75th percentile = 0.156). This means that predicting the RNA velocity (i.e. transcription rate minus the decay rate) for particular individual genes can be challenging, but the combined information is very informative in determining the directionality of cell progression in the trajectory. In terms of velocity correlation, no

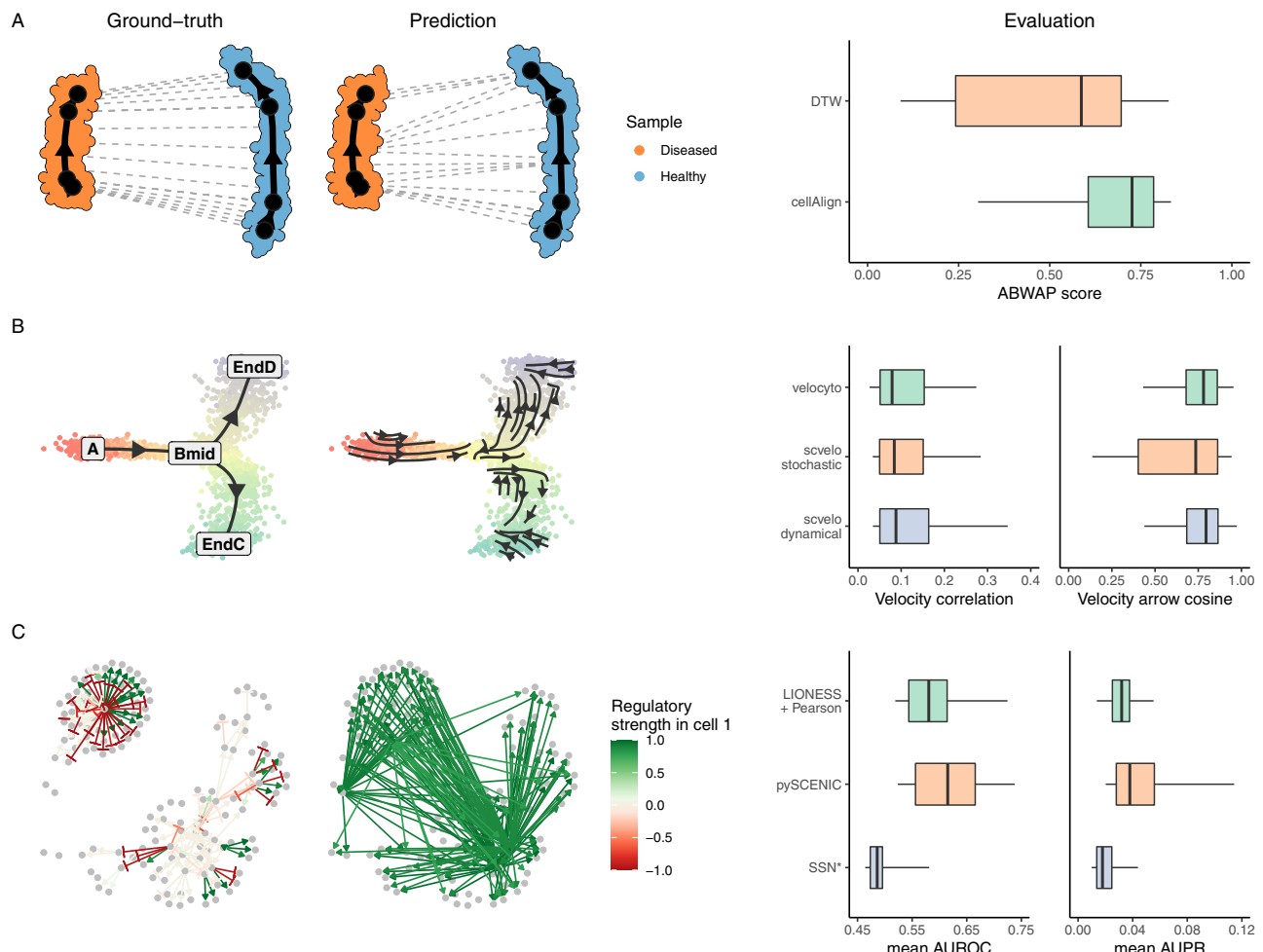

**Fig. 2 dyngen provides ground-truth data for a variety of applications (left), which can be used to quantitatively evaluate methods (right).** Box plots denote the $Q_0$ to $Q_4$ quartile values. **A** Trajectory alignment aligns two trajectories between samples. We evaluate Dynamic Time Warping (DTW) and cellAlign when aligning two linear trajectories with different kinetic parameters based on the area differences between the worst possible alignment and the predicted alignment (Area Between Worst And Prediction, or ABWAP). **B** RNA velocity calculates for each cell the direction in which the expression of each gene is moving. We evaluated scVelo and velocyto by comparing these vectors with the known velocity vector (velocity correlation) and with the known direction of the cellular trajectory in a dimensionality reduction (velocity arrow cosine). **C** Cell-specific network inference (CSNI) predicts the regulatory network of every individual cell. We evaluate each cell-specific regulatory network with typical metrics for network inference: the Area Under the Receiver Operating Characteristics-curve (AUROC) and Area Under the Precision-Recall curve (AUPR). We evaluate three CSNI methods by computing the mean AUROC and AUPR across all cells.

method performed significantly better than the other, whereas "scvelo stochastic" performed slightly worse than "scvelo dynamical" and velocyto in terms of velocity arrow cosine score. Note that, given that some genes are more informative in determining the overall directionality of cell progression, performing a feature selection before computing the embedded dimensionality reduction might result in significantly improved velocity arrow cosine scores.

Use-case "Cell-specific network inference" (CSNI). CSNI methods predict not only which transcription factors regulate which target genes, but also aim to identify how active each interaction is in each of the cells, since interactions can be turned off and on depending on the cellular state. While a few pioneering CSNI approaches have already been developed[32–34], a quantitative assessment of their performance is until now lacking. This is not surprising, as neither real nor in silico datasets of cell-specific or even cell-type-specific interactions exist that are large enough so that it can be used as a ground-truth for evaluating CSNI methods. Extracting the ground-truth dynamic network in

dyngen is straightforward though, given that we can calculate how target gene expression would change without the regulator being present. We used this ground-truth to compare the performance of three CSNI methods (Supplementary Fig. 3): LIONESS[33], SSN[34], and SCENIC[32]. For each dataset, we computed the mean Area Under the Receiver Operating Characteristic-curve (AUROC) and Area Under the Precision-Recall curve (AUPR) scores of the individual cells. Comparing the mean AUROC and AUPR showed that pySCENIC significantly outperforms both LIONESS and SSN, and in turn that LIONESS significantly outperforms SSN. The poor performance of SSN is expected, as its methodology for predicting a cell-specific is simply computing the difference in Pearson correlation values applied to the whole dataset and the whole dataset minus one sample. This strategy performs poorly in large datasets where cell correlations are high, as the removal of one cell will not yield large differences in correlation values and will result in mostly noise. Overall, pySCENIC almost always performs better than LIONESS, except for a few datasets where LIONESS does manage

to obtain a higher AUROC score. However, by using a different internal network inference (e.g. GENIE3[35] or pySCENIC's GRNBoost2[36]) could significantly increase the performance obtained by LIONESS.

## Discussion

dyngen's single-cell simulations can be used to evaluate common single-cell omics computational methods such as clustering, batch correction, trajectory inference, and network inference. However, the framework is flexible enough to be adaptable to a broad range of applications, including methods that integrate clustering, network inference, and trajectory inference. In this respect, dyngen may promote the development of tools in the single-cell field similarly as other simulators have done in the past[5,37]. Additionally, one could anticipate technological developments in single-cell multi-omics. In this way, dyngen allows designing and evaluating the performance and robustness of new types of computational analyses before experimental data becomes available, comparing which experimental protocol is the most cost-effective in producing qualitative and robust results in downstream analysis. One major assumption of dyngen is that cells are assumed to be well-mixed and independent from each other. Subdividing a cell into multiple 2D or 3D subvolumes or allowing cells to exchange molecules, respectively, could pave the way to better study key cellular processes such as cell division, intercellular communication, and migration[38].

## Methods

The workflow to generate in silico single-cell data consists of six main steps (Supplementary Fig. 4).

**Defining the module network**. One of the main processes involved in cellular dynamic processes is gene regulation, where regulatory cascades and feedback loops lead to progressive changes in expression and decision making. The exact way a cell chooses a certain path during its differentiation is still an active research field, although certain models have already emerged and been tested in vivo. One driver of bifurcation is mutual antagonism, where two genes strongly repress each other[39,40], forcing one of the two to become inactive[41]. Such mutual antagonism can be modelled and simulated[42,43]. Although the two-gene model is simple and elegant, the reality is frequently more complex, with multiple genes (grouped into modules) repressing each other[44].

To start a dyngen simulation, the user needs to define a module network. The module network describes how sets of genes regulate each other and is what mainly determines which dynamic processes occur within the simulated cells.

A module network consists of modules connected together by regulatory interactions, which can be either upregulating or downregulating. A module may have basal expression, which means genes in this module will be transcribed without the presence of transcription factor molecules. A module marked as "active during the burn phase" means that this module will be allowed to generate expression of its genes during an initial warm-up phase. At the end of the dyngen process, cells will not be sampled from the burn phase simulations. Interactions between modules have a strength (which is a positive integer) and an effect (+1 for upregulating, −1 for downregulating).

Several examples of module networks are given in Supplementary Fig. 5. A simple chain of modules (where one module upregulates the next) results in a *linear* process. By having the last module repress the first module, the process becomes *cyclic*. Two modules repressing each other is the basis of a *bifurcating* process, though several chains of modules have to be attached in order to achieve progression before and after the bifurcation process. Finally, a *converging* process has a bifurcation occurring during the burn phase, after which any differences in module regulation is removed.

Note that these examples represent the bare minimum in terms of the number of modules used. Using longer chains of modules is typically desired. In addition, the fate decisions made in this example of a bifurcation is reversible, meaning cells can be reprogrammed to go down a different differentiation path. If this effect is undesirable, more safeguards need to be put in place to prevent reprogramming from occurring.

**Generating the gene regulatory network**. The GRN is generated based on the given module network in four main steps (Supplementary Fig. 6).

Step 1, sampling the transcription factors (TF). The TFs are the main drivers of the molecular changes in the simulation. The user provides a backbone and the number of TFs to generate. Each TF is assigned to a module such that each module

has at least $x$ parameters (default $x = 1$). A TF inherits the 'burn' and 'basal expression' from the module it belongs to.

Step 2, generating the TF interactions. Let each TF be regulated according to the interactions in the backbone. These interactions inherit the effect, strength, and independence parameters from the interactions in the backbone. A TF can only be regulated by other TFs or itself.

Step 3, sampling the target subnetwork. A user-defined number of target genes are added to the GRN. Target genes are regulated by a TF or another target gene, but are always downstream of at least one TF. To sample the interactions between target genes, one of the many FANTOM5[45] GRNs is sampled. The currently existing TFs are mapped to regulators in the FANTOM5 GRN. The targets are drawn from the FANTOM5 GRN weighted by their page rank value, to create an induced GRN. For each target, at most $x$ regulators are sampled from the induced FANTOM5 GRN (default $x = 5$). The interactions connecting a target gene and its regulators are added to the GRN.

Step 4, sampling the housekeeping subnetwork. Housekeeping genes are completely separate from any TFs or target genes. A user-defined set of housekeeping genes is also sampled from the FANTOM5 GRN. The interactions of the FANTOM5 GRN are first subsampled such that the maximum in-degree of each gene is $x$ (default $x = 5$). A random gene is sampled and a breadth-first-search is performed to sample the desired number of housekeeping genes.

**Convert gene regulatory network to a set of reactions**. Simulating a cell's GRN makes use of a stochastic framework which tracks the abundance levels of molecules over time in a discrete quantity. For every gene $G$, the abundance levels of three molecules are tracked, namely of corresponding pre-mRNAs, mature mRNAs and proteins, which are represented by the terms $x_G$, $y_G$, and $z_G$ respectively. The GRN defines how a reaction affects the abundance levels of molecules and how likely it will occur. Gibson and Bruck[46] provide a good introduction to modelling gene regulation with stochastic frameworks, on which many of the concepts below are based.

For every gene in the GRN a set of reactions are defined, namely transcription, splicing, translation, and degradation. Each reaction consists of a propensity function—a formula $f(.)$ to calculate the probability $f(.) \times dt$ of it occurring during a time interval $dt$—and the effect—how it will affect the current state if triggered.

The effects of each reaction mimic the respective biological processes (Supplementary Table 2, middle). Transcription of gene $G$ results in the creation of a single pre-mRNA molecule $x_G$. Splicing turns one pre-mRNA $x_G$ into a mature mRNA $x_G$. Translation uses a mature mRNA $y_G$ to produce a protein $z_G$. Pre-mRNA, mRNA, and protein degradation results in the removal of a $x_G$, $y_G$, and $z_G$ molecule, respectively.

The propensity of all reactions except transcription are all linear functions (Supplementary Table 2, right) of the abundance level of some molecule multiplied by a per-gene constant (Supplementary Table 3). The propensity of transcription of a gene $G$ depends on the abundance levels of its TFs. The per-gene and per-interaction constants are based on the median reported production-rates and half-lives of molecules measured of 5000 mammalian genes[47], except that the transcription rate has been amplified by a factor of 10.

The propensity of the transcription of a gene $G$ is inspired by thermodynamic models of gene regulation[48], in which the promoter of $G$ can be bound or unbound by a set of $N$ transcription factors $H_i$. Let $f(z_1, z_2, …, z_N)$ denote the propensity function of $G$, in function of the abundance levels of the transcription factors. The following subsections explain and define the propensity function when $N = 1$, $N = 2$, and finally for an arbitrary $N$.

*Propensity of transcription when N = 1*. In the simplest case when $N = 1$, the promoter can be in one of two states. In state $S_0$, the promoter is not bound by any transcription factors, and in state $S_1$ the promoter is bound by $H_1$. Each state $S_j$ is linked with a relative activation $\alpha_j$, a number between 0 and 1 representing the activity of the promoter at this particular state. The propensity function is thus equal to the expected value of the activity of the promoter multiplied by the pre-mRNA production rate of $G$.

$$f(y_1, y_2, … , y_N) = \text{xpr} \cdot \sum_{j=0}^{2^N - 1} \alpha_j \cdot P(S_j) \tag{1}$$

For $N = 1$, $P(S_1)$ is equal to the Hill equation, where $k_i$ represents the concentration of $H_i$ at half-occupation and $n_i$ represents the Hill coefficient. Typically, $n_i$ is between $[1, 10]$

$$P(S_1) = \frac{y_1^{n_1}}{k_1^{n_1} + y_1^{n_1}} \tag{2}$$

$$= \frac{(y_1/k_1)^{n_1}}{1 + (y_1/k_1)^{n_1}} \tag{3}$$

The Hill equation can be simplified by letting $\nu_i = \left(\frac{y_i}{k_i}\right)^{n_i}$.

$$P(S_1) = \frac{\nu_1}{1 + \nu_1} \tag{4}$$

Since $P(S_0) = 1 - P(S_1)$, the activation function is formulated and simplified as follows.

$$f(y_1) = \text{xpr} \cdot (\alpha_0 \cdot P(S_0) + \alpha_1 \cdot P(S_1)) \quad (5)$$

$$= \text{xpr} \cdot \left( \alpha_0 \cdot \frac{1}{1 + \nu_1} + \alpha_1 \cdot \frac{\nu_1}{1 + \nu_1} \right) \quad (6)$$

$$= \text{xpr} \cdot \frac{\alpha_0 + \alpha_1 \cdot \nu_1}{1 + \nu_1} \quad (7)$$

*Propensity of transcription when* N = 2. When $N = 2$, there are four states $S_j$. The relative activations $\alpha_j$ can be defined such that $H_1$ and $H_2$ are independent (additive) or synergistic (multiplicative). In order to define the propensity of transcription $f(.)$, the Hill equation $P(S_j)$ is extended for two transcription factors.

Let $w_j$ be the numerator of $P(S_j)$, defined as the product of all transcription factors bound in that state:

$$w_0 = 1 \quad (8)$$

$$w_1 = \nu_1 \quad (9)$$

$$w_2 = \nu_2 \quad (10)$$

$$w_3 = \nu_1 \cdot \nu_2 \quad (11)$$

The denominator of $P(S_j)$ is then equal to the sum of all $w_j$. The probability of state $S_j$ is thus defined as:

$$P(S_j) = \frac{w_j}{\sum_{j=0}^{j < 2^N} w_j} \quad (12)$$

$$= \frac{w_j}{1 + \nu_1 + \nu_2 + \nu_1 \cdot \nu_2} \quad (13)$$

$$= \frac{w_j}{\prod_{i=1}^{i \le N} (\nu_i + 1)} \quad (14)$$

Substituting $P(S_j)$ and $w_j$ into $f(.)$ results in the following equation:

$$f(y_1, y_2) = \text{xpr} \cdot \sum_{j=0}^{2^N - 1} \alpha_j \cdot P(S_j) \quad (15)$$

$$= \text{xpr} \cdot \frac{\sum_{j=0}^{2^N - 1} \alpha_j \cdot w_j}{\prod_{i=1}^{i \le N} (\nu_i + 1)} \quad (16)$$

$$= \text{xpr} \cdot \frac{\alpha_0 + \alpha_1 \cdot \nu_1 + \alpha_2 \cdot \nu_2 + \alpha_3 \cdot \nu_1 \cdot \nu_2}{(\nu_1 + 1) \cdot (\nu_2 + 1)} \quad (17)$$

*Propensity of transcription for an arbitrary* N. For an arbitrary $N$, there are $2^N$ states $S_j$. The relative activations $\alpha_j$ can be defined such that $H_1$ and $H_2$ are independent (additive) or synergistic (multiplicative). In order to define the propensity of transcription $f(.)$, the Hill equation $P(S_j)$ is extended for $N$ transcription factors.

Let $w_j$ be the numerator of $P(S_j)$, defined as the product of all transcription factors bound in that state:

$$w_j = \prod_{i=1}^{i \le N} (j \bmod i) = 1 ? \nu_i : 1 \quad (18)$$

The denominator of $P(S_j)$ is then equal to the sum of all $w_j$. The probability of state $S_j$ is thus defined as:

$$P(S_j) = \frac{w_j}{\sum_{j=0}^{j < 2^N} w_j} \quad (19)$$

$$= \frac{w_j}{\prod_{i=1}^{i \le N} (\nu_i + 1)} \quad (20)$$

Substituting $P(S_j)$ into $f(.)$ yields:

$$f(y_1, y_2, \dots, y_N) = \text{xpr} \cdot \sum_{j=0}^{2^N - 1} \alpha_j \cdot P(S_j) \quad (21)$$

$$= \text{xpr} \cdot \frac{\sum_{j=0}^{2^N - 1} \alpha_j \cdot w_j}{\prod_{i=1}^{i \le N} (\nu_i + 1)} \quad (22)$$

*Propensity of transcription for a large* N. For large values of $N$, computing $f(.)$ is practically infeasible as it requires performing $2^N$ summations. In order to greatly simplify $f(.)$, $\alpha_j$ could be defined as 0 when one of the regulators inhibits transcription and 1 otherwise.

$$\alpha_j = \begin{cases} 0 & \text{if } \exists i : j \bmod i = 1 \text{ and } H_i \text{ represses } G \\ 1 & \text{otherwise} \end{cases} \quad (23)$$

Substituting Eq. (23) into Eq. (22) and defining $R = \{1, 2, \dots, N\}$ and $R^+ = \{i | H_i \text{activates} G\}$ yields the simplified propensity function:

$$f(y_1, y_2, \dots, y_N) = \text{xpr} \cdot \frac{\prod_{i \in R^+} (\nu_i + 1)}{\prod_{i \in R} (\nu_i + 1)} \quad (24)$$

*Independence, synergism, and basal expression.* The definition of $\alpha_j$ as in Eq. (23) presents two main limitations. Firstly, since $\alpha_0 = 1$, it is impossible to tweak the propensity of transcription when no transcription factors are bound. Secondly, it is not possible to tweak the independence and synergism of multiple regulators.

Let ba $\in [0, 1]$ denote the basal expression strength $G$ (i.e. how much will $G$ be expressed when no transcription factors are bound), and sy $\in [0, 1]$ denote the synergism of regulators $H_i$ of $G$, the transcription propensity becomes:

$$f(y_1, y_2, \dots, y_N) = \text{xpr} \cdot \frac{\text{ba} - \text{sy}^{|R^+|} + \prod_{i \in R^+} (\nu_i + \text{sy})}{\prod_{i \in R} (\nu_i + 1)} \quad (25)$$

**Simulate single cells.** dyngen uses Gillespie's SSA[12] to simulate dynamic processes. An SSA simulation is an iterative process where at each iteration one reaction is triggered.

Each reaction consists of its propensity—a formula to calculate the probability of the reaction occurring during an infinitesimal time interval—and the effect—how it will affect the current state if triggered. Each time a reaction is triggered, the simulation time is incremented by $\tau = \frac{1}{\sum_j \text{prop}_j} \ln \left( \frac{1}{r} \right)$, with $r \in U(0, 1)$ and $\text{prop}_j$ the propensity value of the $j$th reaction for the current state of the simulation.

GillespieSSA2 is an optimised library for performing SSA simulations. The propensity functions are compiled to C++ and SSA approximations can be used which allow triggering many reactions simultaneously at each iteration. The framework also allows storing the abundance levels of molecules only after a specific interval has passed since the previous census. By setting the census interval to 0, the whole simulation's trajectory is retained but many of these time points will contain very similar information. In addition to the abundance levels, also the propensity values and the number of firings of each of the reactions at each of the time steps can be retained.

**Simulate experiment.** From the SSA simulation we obtain the abundance levels of all the molecules at every state. We need to replicate technical effects introduced by experimental protocols in order to obtain data that is similar to real data. For this, the cells are sampled from the simulations and molecules are sampled for each of the cells. Gene capture rates and library sizes are empirically derived from real datasets to match real technical variation.

*Sample cells.* In this step, $N$ cells are sampled from the simulations. Two approaches are implemented: sampling from an unsynchronised population of single cells (snapshot) or sampling at multiple time points in a synchronised population (time series).

Snapshot. The backbone consists of several states linked together by transition edges with length $L_i$, to which the different states in the different simulations have been mapped (Supplementary Fig. 7A). From each transition, $N_i = N / \frac{L_i}{\sum L_i}$ cells are sampled uniformly, rounded such that $\sum N_i = N$.

Time series. Assuming that the final time of the simulation is $T$, the interval [0, $T$] is divided into $k$ equal intervals of width $w$ separated by $k - 1$ gaps of width $g$. $N_i = N / k$ cells are sampled uniformly from each interval (Supplementary Fig. 7B), rounded such that $\sum N_i = N$. By default, $k = 8$ and $g = 0.75$. For usual dyngen simulations, $10 \le T \le 20$. For larger values of $T$, $k$ and $g$ should be increased accordingly.

*Sample molecules.* Molecules are sampled from the simulation to replicate how molecules are experimentally sampled. A real dataset is downloaded from a repository of single-cell RNA-seq datasets[49]. For each in silico cell $i$, draw its library size $ls_i$ from the distribution of transcript counts per cell in the real dataset. The capture rate $cr_j$ of each in silico molecule type $j$ is drawn from $N(1, 0.05)$. Finally, for each cell $i$, draw $ls_i$ molecules from the multinomial distribution with probabilities $cr_j \times ab_{i,j}$ with $ab_{i,j}$ the molecule abundance level of molecule $j$ in cell $i$.

*Comparison between a dyngen and a reference dataset.* Comparison between a dyngen dataset and the reference dataset it used in terms of characteristic single-cell omics features showed that dyngen produces datasets with highly similar data characteristics (Supplementary Note 1). Supplementary Note 1 was generated using countsimQC[50].

**Simulating batch effects.** Simulating batch effects can be performed in multiple ways. One such way is to perform the first two steps of the creation of a dyngen model (defining the module network and generating the GRN). For each desired batch, create a separate model for which random kinetics are generated and perform all subsequent dyngen steps (convert to reactions, simulate gold standard, simulate single cells, simulate experiment). Since each separate model has different underlying kinetics, the combined output will resemble having batch effects.

**Determining the ground-truth trajectory**. To construct the ground-truth trajectory, the user needs to provide the ground-truth state network alongside the initial module network (Supplementary Fig. 8). Each edge in the state network specifies which modules are allowed to change in expression in transitioning from one state to another. For each edge, a simulation is run using the end state of an upstream branch as the initial expression vector, and only allowing the modules as predefined by the attribute to change.

As an example, consider the cyclic trajectory shown in Supplementary Fig. 8. State S0 begins with an expression vector of all zero values. To simulate the transition from S0 to S1, regulation of the genes in modules A, B, and C are turned on. After a predefined period of time, the end state of this transition is considered the expression vector of state S1. To simulate the transition from S1 to S2, regulation of the genes in modules D and E are turned on, while the regulation of genes in module C is turned off. During this simulation, the expression of genes in modules A, B, D, and E is thus allowed to change. The end state of the simulation is considered the expression vector of state S2.

For each of the branches in the state network, an expression matrix and the corresponding progression time along that branch are retained. To map a simulated cell to the ground-truth, the correlation between its expression values and the expression matrix of the ground-truth trajectory is calculated, and the cell is mapped to the position in the ground-truth trajectory that has the highest correlation.

**Determining the cell-specific ground-truth regulatory network**. Calculating the regulatory effect of a regulator $R$ on a target $T$ (Supplementary Fig. 4F) requires determining the contribution of $R$ in the propensity function of the transcription of $T$ with respect to other regulators. This information is useful, amongst others, for benchmarking cell-specific network inference methods.

The regulatory effect of $R$ on $T$ at a particular state $S$ is defined as the change in the propensity of transcription when $R$ is set to zero, scaled by the inverse of the pre-mRNA production rate of $T$. More formally:

$$\text{regeffect}_G = \frac{\text{proptrans}_G(S) - \text{proptrans}_G(S[z_T \leftarrow 0])}{\text{xpr}_G} \qquad (26)$$

Determining the regulatory effect for all interactions and cells in the dataset yields the complete cell-specific ground-truth GRN. The regulatory effect lies between $[-1, 1]$, where $-1$ represents complete inhibition of $T$ by $R$, 1 represents maximal activation of $T$ by $R$, and 0 represents inactivity of the regulatory interaction between $R$ and $T$.

**Comparison of cell-specific network inference methods**. 42 datasets were generated using the 14 different predefined backbones and three different seeds. For every cell in the dataset, the transcriptomics profile and the corresponding cell-specific ground-truth regulatory network was determined.

We selected three cell-specific NI methods: SCENIC[32], LIONESS[33,51], and SSN[34].

LIONESS[33] runs a NI method multiple times to construct cell-specific GRNs. LIONESS first infers a GRN with all of the samples. A second GRN is inferred with all samples except one particular profile. The cell-specific GRN for that particular profile is defined as the difference between the two GRN matrices. This process is repeated for all profiles, resulting in a cell-specific GRN. By default, LIONESS uses PANDA[52] to infer GRNs, but since dyngen does not produce motif data and motif data is required by PANDA, PANDA is inapplicable in this context. Instead, we used the lionessR[53] implementation of LIONESS, which uses by default the Pearson correlation as a NI method. We marked results from this implementation as "LIONESS + Pearson".

SSN[34] follows, in essence, the exact same methodology as LIONESS except that it specifically only uses the Pearson correlation. It is worth noting that the LIONESS preprint was released before the publication of SSN. Since no implementation was provided by the authors, we implemented SSN in R using basic R and tidyverse functions[54] and marked results from this implementation as "SSN*".

SCENIC[32] consists of four main steps. First, classical network inference is performed with stochastic gradient boosting machines using `arboreto`[36]. Second, the top 10 regulators of every target gene are selected. Interactions are grouped together in 'modules'; each module contains one regulator and all of its targets. Next, the modules are filtered using motif analysis. Finally, for each module and each cell, an activity score is calculated using AUCell. As a post-processing of this output, all modules and the corresponding activity scores are combined back into a cell-specific GRN consisting of (cell, regulator, target, score) pairs. For this analysis, the Python implementation of SCENIC was used, namely pySCENIC[55]. Since dyngen does not generate motif data, step 3 in this analysis is skipped.

The AUROC and AUPR metrics are common metrics for evaluating a predicted GRN with a ground-truth GRN[56]. To compare a predicted cell-specific GRN with the ground-truth cell-specific GRN, the top 10,000 interactions per cell is retained, and the mean AUROC and AUPR scores are calculated across all cells.

We compared the mean AUROC and AUPR scores obtained by the three CSNI methods across all datasets by performing pairwise non-parametric paired two-sided Durbin–Conover tests[57] using `pairwiseComparisons`[58]. Test statistics and $p$ values for the all pairwise combinations are reported in the Source Data file. Reported $p$ values are adjusted for multiple testing using Holm correction[59].

**Comparison of RNA velocity methods**. Three datasets were generated for each of the 14 different predefined backbones, resulting in a collection of 42 datasets. Throughout each of the simulation, the propensity of the transcription and mRNA decay is collected, as the RNA velocity of a gene at any point in the simulation is the difference between the transcription propensity and the mRNA decay propensity.

We applied two RNA velocity methods: velocyto[30], as implemented in the `velocyto.py` package, and scvelo method[31], as implemented in the `scvelo` package. For scvelo, we chose two parameter settings for "mode", namely "stochastic" and "dynamical". For both methods, we used the same normalised data as provided by dyngen, with no extra cell or feature filtering, but otherwise matched the parameters to their respective tutorial vignettes as well as possible.

We compared each RNA velocity prediction to the ground-truth using two metrics: the velocity correlation and the velocity arrow cosine. For the velocity correlation, we extracted a ground truth RNA velocity by subtracting for each mRNA molecule the propensity of its production by the propensity of its degradation. If the expression of an mRNA will increase in the future, this value is positive, while it is negative if it is going to decrease. For each gene, we determined its velocity correlation by calculating the Spearman rank correlation between the ground truth velocity and the observed velocity. For the velocity arrow cosine, we determined a set of 100 trajectory waypoints uniformly spread on the trajectory. For each waypoint, we weighted each cell based on a Gaussian kernel on its geodesic distance from the waypoint. These weights were used to calculate a weighted average velocity vector of each waypoint. We then calculated for each waypoint the cosine similarity between this velocity vector and the known direction of the trajectory.

We compared the velocity correlation and velocity arrow cosine scores obtained by velocyto and scvelo across all datasets by performing pairwise non-parametric paired two-sided Durbin–Conover tests[57] using `pairwiseComparisons`[58]. Test statistics and $p$ values for the all pairwise combinations are reported in the Source Data file. Reported $p$ values are adjusted for multiple testing using Holm correction[59].

**Comparison of trajectory alignment**. Four custom linear backbones of varying sizes were constructed. For each of these backbones, 10 datasets were generated with 10 different seeds, resulting in a total of 40 datasets. Every dataset is generated in three main steps. First, the GRN is generated based on the given backbone. Next, generating the kinetics, gold standard, and cells is performed twice, resulting in two sub-datasets. Finally, the two sub-datasets are combined and cells are sampled from the combined dataset. Since the two sub-datasets were simulated with different kinetic parameters, the combined dataset will contain two trajectories.

On each combined dataset we applied two trajectory alignment methods, DTW[24] and cellAlign[28]. DTW is designed to align temporal sequences by dilating or contracting the sequences to best match each other. cellAlign uses DTW to perform this alignment, but first interpolates and rescales the input data in order to better cope with single-cell omics data.

To evaluate a trajectory alignment method on a combined dataset we computed the geodesic distances of each cell from the start of the trajectory, also called the *pseudotime*. For each dataset, the pseudotime values are rescaled between 0 and 1 to allow for easier comparison. A trajectory alignment produces a sequence of index pairs $[(i_0, j_0), (i_1, j_1), …, (i_N, j_N)]$, where $i_0$ and $j_0$ are equal to 0 (the first position in both pseudotime series), $i_N$ and $j_N$ are equal to the respective last positions in the pair of pseudotime series, and $[i_0, i_1, …, i_N]$ and $[j_0, j_1, …, j_N]$ are in ascending order and can contain duplicates values. The ABWAP metric is defined as follows, where $pt_1$ and $pt_2$ are the unit pseudotime vectors. See Supplementary Fig. 1D for a visual interpretation of this metric.

$$\text{ABWAP} = 1 - \text{area\_under\_curve}(pt_1[i_0..i_N] + pt_2[j_0..j_N], abs(pt_1[i_0..i_N] - pt_2[j_0..j_N])) \qquad (27)$$

We compared the ABWAP scores obtained by DTW and cellAlign across all datasets by performing pairwise non-parametric paired two-sided Durbin–Conover tests[57] using `pairwiseComparisons`[58]. Test statistics and $p$ values for the all pairwise combinations are reported in the Source Data file. Reported $p$ values are adjusted for multiple testing using Holm correction[59].

**Comparison of scalability and runtime**. Simulating a bifurcating cycle dataset with 10,000 genes and 10,000 required in total 1147 s (Supplementary Note 2). Fixing the number of genes and varying the number of cells showed that the execution time of dyngen scales linearly w.r.t. the number of cells (Supplementary Note 2). Fixing the number of cells and varying the number of genes also showed that the execution time of dyngen scales linearly w.r.t. the number of genes (Supplementary Note 2). These timings were measured using 30 (out of 32) threads using a AMD Ryzen 9 5950X clocked at 3.4GHz.

**Reporting summary**. Further information on research design is available in the Nature Research Reporting Summary linked to this article.

## Data availability

Source data for box plots in Fig. 2 and Supplementary Figs. 1–3 are provided with this paper. All code and data required to reproduce the analysis are available on GitHub at https://github.com/dynverse/dyngen_manuscript. The datasets generated for the different use cases are available on Zenodo with record number 4637926 (https://doi.org/10.5281/zenodo.4637926). Source data are provided with this paper.

## Code availability

Results in this manuscript were generated with R 4.0.3 and dyngen 1.0.0. dyngen is available as an open-source software package at https://cran.r-project.org/package=dyngen and also on Zenodo with record number 4751443 (https://doi.org/10.5281/zenodo.4751443). The analyses performed in this manuscript are available on GitHub at https://github.com/dynverse/dyngen_manuscript. The version numbers of downstream dependencies of dyngen and dyngen_manuscript used in this study are: anndata 0.7.5.1, assertthat 0.2.1, babelwhale 1.0.1, bit 4.0.4, bit64 4.0.5, carrier 0.1.0, cellAlign 0.1.0, codetools 0.2-18, colorspace 2.0-0, compiler 4.0.4, crayon 1.4.1.9000, data. table 1.13.4, DBI 1.1.1, debugme 1.1.0, desc 1.2.0, digest 0.6.27, dplyr 1.0.5, dtw 1.22-3, dynparam 1.0.1, dynutils 1.0.6, dynwrap 1.2.2, ellipsis 0.3.1, fansi 0.4.2, farver 2.1.0, future 1.20.1, future.apply 1.7.0, generics 0.1.0, ggforce 0.3.2, ggplot2 3.3.3, ggraph 2.0.4, ggrepel 0.9.0, GillespieSSA2 0.2.7, globals 0.14.0, glue 1.4.2, graphlayouts 0.7.1, grid 4.0.4, gridExtra 2.3, gtable 0.3.0, gtools 3.8.2, hdf5r 1.3.3, hms 1.0.0, igraph 1.2.6, irlba 2.3.3, jsonlite 1.7.2, lattice 0.20-41, lifecycle 1.0.0, lisi 1.0.0, listenv 0.8.0, lmds 0.1.0, magrittr 2.0.1, MASS 7.3-53, Matrix 1.3-2, matrixStats 0.57.0, munsell 0.5.0, parallel 4.0.4, parallelly 1.21.0, patchwork 1.1.1, pbapply 1.4-3, pheatmap 1.0.12, pillar 1.5.1, pkgconfig 2.0.3, plyr 1.8.6, polyclip 1.10-0, pracma 2.2.9, processx 3.4.5, proxy 0.4-24, proxyC 0.1.5, ps 1.6.0, purrr 0.3.4, R6 2.5.0, RANN 2.6.1, rappdirs 0.3.3, RColorBrewer 1.1-2, Rcpp 1.0.6, RcppParallel 5.0.2, RcppXPtrUtils 0.1.1, readr 1.4.0, remotes 2.2.0, reshape2 1.4.4, reticulate 1.18-9007, rlang 0.4.10, rprojroot 2.0.2, scales 1.1.1, sctransform 0.3.2, scvelo 0.1.0.9000, stringi 1.5.3, stringr 1.4.0, tibble 3.0.5, tidygraph 1.2.0, tidyr 1.1.2, tidyselect 1.1.0, tools 4.0.4, tweenr 1.0.1, utf8 1.1.4, vctrs 0.3.6, viridis 0.5.1, viridisLite 0.3.0, yaml 2.2.1.

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

## Acknowledgements

This project has been made possible in part by grant number 2020-218899 from the Chan Zuckerberg Initiative DAF, an advised fund of Silicon Valley Community Foundation. This research received funding from the Flemish Government under the "Onderzoeksprogramma Artificiële Intelligentie (AI) Vlaanderen" program.

## Author contributions

W.S. and R.C. designed the study. R.C., W.S., and L.D. performed the experiments and analysed the data. R.C. and W.S. implemented the dyngen software package. R.C., W.S., L.D., and Y.S. wrote the manuscript. Y.S. supervised the project.

## Competing interests

The authors declare no competing interests.
