## [Peer Review File · Nature Communications]

REVIEWER COMMENTS

Reviewer #2 (Remarks to the Author):

Remarks to the Author (NCOMMS-20-25289-T)

Cannoodt and Saelens et al. introduce a software package for the simulation of multi-modal scRNA-seq data. The underlying model includes most relevant dynamic processes to gene expression measurements. As opposed to other simulation engines, it captures all stages of the transcription process, is based on a user-defined gene module network, and is capable of generating complex scenarios. The authors demonstrate its usability by evaluating computational tools for trajectory alignment, network inference and RNA velocity.

It could be of great benefit to the community and should enable new benchmarking efforts and the development of new computational methods. Overall the paper is well assembled, nicely concise with good supplement, and has insightful figures - however some major points on simulation method evaluation, applications and benchmarking are missing IMHO, see below. I am convinced that with these additions, this could be a great paper for Nat Comm.

Major comments:

1. Bioinformatics has been looking into GRNs, their simulation and estimation for more than a decade (in bulk at the time), and I miss some important references and insights from then. Such as alternate definitions of what a GRN is - eg does logic play a role, should you use boolean networks (see Albert&Othma, JTB 2003) -, do old-school bulk simulators such as GeneNetWeaver (Bioinformatics 2011) be used, can one use transformation methods (eg Wittmann et al, BMC Sysbio 2009) and how is this different from what has been introduced here (what is novelty here).
2. Similarly, the single cell genomics field has used simulations also for dynamics for some time as evaluation at least implicitly. Please discuss advances and differences or similarities to those, eg. to simulations from (Ocone, Bioinf 2015) or (Prapta, Nat Meth 2020).
3. One of the main aims of any simulation framework should be to capture features of real biological data. I'd be great if the authors could demonstrate the fidelity of the generated data by comparing it to some real datasets, e.g. distribution of mean gene expression and number of zero counts per gene. How does it perform compared to other simulation frameworks? This would help me as well as community to evaluate how realistic the induced gene correlations/dependencies are, and what may be missing.
4. Simulated data for the described key applications of dynngen (network inference, trajectory alignment, and RNA velocity) has already been generated within the respective works of lioness (Fig. S1+S4), cellAlign (Fig. S6), velocityto (Fig. S5), scvelo (Fig. S3), dynamo (Fig. 1), many of which also use Gillespie. What does dynngen add conceptually?
5. The authors evaluate several computational tools, including the afore-mentioned. Even if the authors mention not to aim for a large-scale benchmarking, they still should provide some insights into why some tools perform better than others. Having an interpretable simulation engine in hand, it is unfortunate that it yet lacks any explanation and the reader is left alone with performance statements that are not well-founded. Some intuition would be very desirable. Why does SCENIC outperform the other methods? Why does smoothing improve trajectory alignment? And what causes differences in velocity estimation? Could you, for instance, generate an example where the differences in velocity estimates become more obvious, where one estimation procedure fails to fully capture the known direction?

Minor comments:

1. Could the authors please discuss runtime and scalability, which obviously can easily become a weakness for such a complex model.
2. Please state how many genes have been used in the examples. It seems like the number of genes used in the examples is rather small (< 100). Wouldn't it be better to use a more realistic number of features (~10,000)?
3. Congratulations for making the clearly documented package available. Installation and going through the vignette works fine. At several points during the simulation files are downloaded without asking permission. It may be better practice to ask the user for permission, or download into a temporary directory, or read the files directly without downloading, or include in the package etc. Further, it might be very useful to output the simulated data to a standard object, such as Seurat or SingleCellExperiment, or to provide instructions on how to convert it.

Response to reviewers' comments

dyngen rebuttal

We would like to thank the reviewer for the constructive comments, which has motivated us to greatly improve the experiments performed in this work and the documentation provided for the tool. Notable changes are:

- We simplified and reran all use-case experiments, which allowed us to add some interpretation of the results for each of the use cases.
- We performed new experiments assessing the scalability of dyngen and the similarity in output in comparison to a reference (real) dataset. The results are included in the supplementary material.
- We created a documentation website available at dyngen.dynverse.org, which contains not only the documentation on how the package can be used, but also material created based on suggestions by the reviewer, namely expanded installation instructions and experiments on scalability and comparison to reference datasets.
- We added conversion functions for outputting the dyngen output to one of the many commonly used single-cell omics formats: `anndata`, `dyno`, `SingleCellExperiment`, `Seurat`.

Comment 1

Bioinformatics has been looking into GRNs, their simulation and estimation for more than a decade (in bulk at the time), and I miss some important references and insights from then. Such as alternate definitions of what a GRN is - eg does logic play a role, should you use boolean networks (see Albert&Othma, JTB 2003) -, do old-school bulk simulators such as GeneNetWeaver (Bioinformatics 2011) be used, can one use transformation methods (eg Wittmann et al, BMC Sysbio 2009) and how is this different from what has been introduced here (what is novelty here).

Certainly simulating expression datasets using GRNs has been around for a long time and dyngen is inspired heavily by such software (GeneNetWeaver in particular). Nevertheless, there are great differences between bulk and single-cell simulators: the data characteristics of single-cell omics are vastly different from bulk omics, typically having much lower library sizes and a higher dropout rate, but also a high number of profiles. The low library sizes, in particular, are problematic as ODEs are ill-suited for performing low-molecule simulations. This necessitates the development of new single-cell simulators.

We have added paragraph §3 to the introduction to discuss the relevance of bulk simulators and why they are not suitable for simulating single-cell expression data:

Simulating realistic data requires that the underlying biology is recapitulated as best as possible, and in the case of transcriptomics data this typically involves modelling the underlying gene regulatory networks. Simulators of "bulk" microarray or RNA-sequencing profiles simulate biological processes (e.g. transcription, translation) by translating a database of known regulatory interactions into a set of ordinary differential equations (ODE) [1, 2, 3, 4]. These methods have been instrumental in performing benchmarking studies [5, 6, 7]. However, the advent of single-cell omics introduced several new types of analyses (e.g. trajectory inference, RNA velocity, cell-specific network inference) which exploit the higher resolution of single-cell versus bulk omics [8]. In addition, the data characteristics of single-cell omics are vastly different from bulk omics, typically having much lower library sizes and a higher dropout rate, but also a high number of profiles [9]. The low library sizes, in particular, are problematic as ODEs are ill-suited for performing low-molecule simulations [10]. This necessitates the development of new single-cell simulators.

Comment 2

Similarly, the single cell genomics field has used simulations also for dynamics for some time as evaluation at least implicitly. Please discuss advances and differences or similarities to those, eg. to simulations from (Ocone, Bioinf 2015) or (Pratapa, Nat Meth 2020).

We expanded upon paragraph §4 and made a more direct comparison to the methods listed in Supplemental Table 1. Paragraph §4:

To this end, single-cell omics simulators emulate the technical procedures from single-cell omics protocols. Simulators such as Splatter [11], powsimR [12], PROSSTT [13] and SymSim [14]) have already been widely used to compare single-cell methods [15, 16, 17, 18] and perform independent benchmarks [19, 20, 21]. However, by focusing more on simulating the single-cell omics protocol (e.g. RNA capture, amplification, sequencing) and less on the underlying biology (e.g. transcription, splicing, translation), their applicability and reusability is limited towards the specific application for which they were designed (e.g. benchmarking clustering or differential expression methods), and extending these tools to include additional modalities or experimental conditions is challenging.

We did not include the reference to Ocone Bioinf 2015 as the advances proposed by this research were mainly related to performing Network Inference. In terms of generating synthetic data, the simulator used by Ocone et al. offered little advantage over aforementioned bulk simulators. We also did not include a reference to Pratapa et al., as their methodology is borrowed heavily from dyngen itself and offers little advantage over it.

Comment 3

One of the main aims of any simulation framework should be to capture features of real biological data. I'd be great if the authors could demonstrate the fidelity of the generated data by comparing it to some real datasets, e.g. distribution of mean gene expression and number of zero counts per gene. How does it perform compared to other simulation frameworks? This would help me as well as community to evaluate how realistic the induced gene correlations/dependencies are, and what may be missing.

A vignette has been added to the package and supplementary material which uses countsimQC [22] to explore the characteristics of an dyngen simulation and the reference dataset used by it (dynverse.org/articles/advanced_topics/comparison_reference.html). The distribution plots in this vignette, while not perfect, nicely demonstrates that the output generated by dyngen mimics the given reference dataset for the evaluated metrics.

Comment 4

Simulated data for the described key applications of dyngen (network inference, trajectory alignment, and RNA velocity) has already been generated within the respective works of lioness (Fig. S1+S4), cellAlign (Fig. S6), velocityto (Fig. S5), scvelo (Fig. S3), dynamo (Fig. 1), many of which also use Gillespie. What does dyngen add conceptually?

We modified paragraphs §5 and §6 to better highlight the scientific contributions dyngen makes:

We introduce dyngen, a method for simulating cellular dynamics at a single-cell, single-transcript resolution (Figure 1). This problem is tackled in three fully-configurable main steps. First, biological processes are mimicked by translating a gene regulatory network in a set of reactions (regulation, transcription, splicing, translation). Second, individual cells are simulated using Gillespie's stochastic simulation algorithm [10], which is designed to work well in low-molecule simulations. Finally, real reference datasets are used to emulate single-cell omics profiling protocols.

Throughout a simulation, dyngen tracks many layers of information, including the abundance of any molecule in the cell, the progression of the cell along a dynamic process, and the activation strength of individual regulatory interactions. In addition, dyngen can simulate a large variety of dynamic processes (e.g. cyclic, branching, disconnected) as well as a broad range of experimental conditions (e.g. batch effects and time-series, perturbation

and single-cell knockdown experiments). For these reasons, *dyngen* can cater to a wide range of benchmarking applications, including trajectory inference, trajectory alignment, and trajectory differential expression (Table S1).

To the best of our knowledge, the works listed above use very different approaches to benchmarking their tool:

- LIONESS, network inference method: Uses randomly generated networks & boolean networks to generate synthetic data. The networks used to generate datasets are very synthetic and completely devoid of any biological relevance. By using Gillespie's SSA algorithm instead of boolean networks, we can use *dyngen* to track many layers of information (e.g. the number of times a biological reaction has been triggered during a given time interval). This allows us to benchmark methods which require this information (e.g. RNA velocity).
- cellAlign, trajectory alignment: Perturbs real datasets to assess the robustness of cellAlign. When benchmarking a tool, assessing its robustness is almost as important as assessing its accuracy. Robustness analysis can be performed without knowing the gold standard, whereas the gold standard is needed for assessing the accuracy of a tool.
- velocity, RNA velocity: Uses ODEs to simulate splicing mechanics. The code for reproducing these datasets couldn't be found. Based on the description, these simulations were limited to simulating only the splicing mechanics under a particular set of conditions, thereby limiting its applicability outside of this particular context.
- scvelo, RNA velocity: Same as velocity.
- dynamo, RNA velocity: Uses Gillespie's SSA to simulate splicing reactions. However, this system is very limited, including only two genes. This application of Gillespie's SSA seems very tailored towards this particular use-case and would require major redesigns to be made applicable elsewhere.

It should be noted that from the list above, only LIONESS really compares its accuracy performance with other methods using a quantitative metric. We would thus argue that these cases further highlight the need for broadly applicable synthetic data generators such as *dyngen*.

Comment 5

The authors evaluate several computational tools, including the afore-mentioned. Even if the authors mention not to aim for a large-scale benchmarking, they still should provide some insights into why some tools perform better than others. Having an interpretable simulation engine in hand, it is unfortunate that it yet lacks any explanation and the reader is left alone with performance statements that are not well-founded. Some intuition would be very desirable. Why does SCENIC outperform the other methods? Why does smoothing improve trajectory alignment? And what causes differences in velocity estimation? Could you, for instance, generate an example where the differences in velocity estimates become more obvious, where one estimation procedure fails to fully capture the known direction?

Performing in-depth benchmarks on each of these use-cases can be considered separate studies on their own and is beyond the scope of this work. However, we did reanalyse the experiments and added interpretation to the respective sections.

Trajectory alignment

We removed DTW+smoothing (as it was a custom method developed for the sake of this analysis) and added cellAlign to the comparison, as cellAlign also uses smoothing and DTW to get to the same objective, and has been developed specifically with single cell data in mind.

Interpretation added to manuscript:

Overall, cellAlign performs significantly better than DTW (Figure S1, which is likely due to the interpolation and scaling steps provided by cellAlign, reducing noise in the data and improving the comparability of the trajectories. Note that, in this comparison, only linear trajectory alignment is performed. While *dyngen* can generate non-linear trajectories

(e.g. cyclic or branching), both aligning non-linear trajectories and constructing a quantitative accuracy metric for non-linear trajectory alignment is not trivial and an avenue for future work.

RNA velocity

We removed some parameter settings which are designed to be very similar and thus have no added benefit in the visualisation. We removed the 'difficulty' settings for the different datasets, as the 'easy' and 'medium' datasets were unrealistic; only the 'hard' settings were retained. While there are no large difference in performance between the different methods, we added interpretation on why methods obtain low scores on one metric and high scores on the other.

Interpretation added to manuscript:

While both velocityto and scvelo obtained high scores for the velocity arrow cosine metric (overall 25th percentile = 0.606), the velocity correlation is rather low (overall 75th percentile = 0.156). This means that predicting the RNA velocity (i.e. transcription rate minus the decay rate) for any particular gene is challenging, but the combined information is very informative in determining the directionality of cell progression in the trajectory. In terms of velocity correlation, no method performed significantly better than the other, whereas "scvelo stochastic" performed slightly worse than "scvelo dynamical" and velocityto in terms of velocity arrow cosine score.

Cell-specific network inference

We added interpretation on why certain methods significantly outperform others and provide possible solutions to problems, where possible.

Interpretation added to manuscript:

Comparing the mean AUROC and AUPR showed that pySCENIC significantly outperforms both LIONESS and SSN, and in turn that LIONESS significantly outperforms SSN. The poor performance of SSN is expected, as its methodology for predicting a cell-specific is simply computing the difference in Pearson correlation values applied to the whole dataset and the whole dataset minus one sample. This strategy performs poorly in large datasets where cell correlations are high, as the removal of one cell will not yield large differences in correlation values and will result in mostly noise. Overall, pySCENIC almost always performs better than LIONESS, except for a few datasets where LIONESS does manage to obtain a higher AUROC score. However, by using a different internal network inference (e.g. GENIE3 [23] or pySCENIC's GRNBoost2 [24]) could significantly increase the performance obtained by LIONESS.

Minor comment 1

Could the authors please discuss runtime and scalability, which obviously can easily become a weakness for such a complex model.

A vignette has been added to the package and supplementary material explores the execution time of dyngen for an increasing number of cells and genes (dyngen.dynverse.org/articles/advanced_topics/scalability_and_performance/). These figures show that the execution time of dyngen scales linearly w.r.t. the number of cells and also linearly w.r.t. the number of genes.

Minor comment 2

Please state how many genes have been used in the examples. It seems like the number of genes used in the examples is rather small (< 100). Wouldn't it be better to use a more realistic number of features (~10,000)?

Unfortunately, the examples shown in the package documentation need to be very small to abide by CRAN package submission policies, as the complete testing of the software package (including

but not limited to running all examples) can strictly take 10 minutes on CRAN's server using only one thread.

However, the scalability vignette mentioned above also shows the execution time of each of the steps taken by `dyngen` in simulating a 10'000 by 10'000 dataset, which takes about 20 minutes to complete.

Minor comment 3

Congratulations for making the clearly documented package available. Installation and going through the vignette works fine. At several points during the simulation files are downloaded without asking permission. It may be better practice to ask the user for permission, or download into a temporary directory, or read the files directly without downloading, or include in the package etc.

By default, required files are downloaded to a temporary directory. Unfortunately, these files cannot be included inside the package as they are reference datasets which are much larger than what is allowed by CRAN policies.

Based on the reviewers suggestion, we modified the installation instructions such that it better reflects that files will be downloaded when using `dyngen`, and provides instructions on how to configure the system such that it will cache these files in a user-specified directory (dynverse.org/articles/installation.html).

Minor comment 4

Further, it might be very useful to output the simulated data to a standard object, such as `Seurat` or `SingleCellExperiment`, or to provide instructions on how to convert it.

Thank you for this suggestion. We agree that conversion functions from the typical `dyngen` output to the typical object types familiar to single-cell omics programmers would be very useful. Therefore, we included such functions in the latest version of `dyngen` to convert its output to the following formats: `anndata`, `dyno`, `SingleCellExperiment`, `Seurat`.

References

- [1] Sushmita Roy, Margaret Werner-Washburne, and Terran Lane. "A System for Generating Transcription Regulatory Networks with Combinatorial Control of Transcription". In: *Bioinformatics* 24.10 (May 15, 2008), pp. 1318–1320. ISSN: 1367-4803. DOI: 10.1093/bioinformatics/btn126.
- [2] Hendrik Hache et al. "GeNGe: Systematic Generation of Gene Regulatory Networks". In: *Bioinformatics* 25.9 (May 1, 2009), pp. 1205–1207. ISSN: 1367-4803. DOI: 10.1093/bioinformatics/btp115.
- [3] Thomas Schaffter, Daniel Marbach, and Dario Floreano. "GeneNetWeaver: In Silico Benchmark Generation and Performance Profiling of Network Inference Methods." In: *Bioinformatics* 27.16 (Aug. 2011), pp. 2263–2270. ISSN: 1367-4811. DOI: 10.1093/bioinformatics/btr373. pmid: 21697125.
- [4] Tim Van den Bulcke et al. "SynTReN: A Generator of Synthetic Gene Expression Data for Design and Analysis of Structure Learning Algorithms". In: *BMC Bioinformatics* 7.1 (Jan. 26, 2006), p. 43. ISSN: 1471-2105. DOI: 10.1186/1471-2105-7-43.
- [5] Robert J Prill et al. "Towards a Rigorous Assessment of Systems Biology Models: The DREAM3 Challenges". In: *PLoS ONE* 5.2 (Jan. 2010), e9202. ISSN: 1932-6203. DOI: 10.1371/journal.pone.0009202. pmid: 20186320.
- [6] Daniel Marbach et al. "Revealing Strengths and Weaknesses of Methods for Gene Network Inference". In: *Proceedings of the National Academy of Sciences* 107.14 (Apr. 2010), pp. 6286–6291. ISSN: 1091-6490. DOI: 10.1073/pnas.0913357107. pmid: 20308593.
- [7] Daniel Marbach et al. "Wisdom of Crowds for Robust Gene Network Inference". In: *Nature methods* 9.8 (July 2012), pp. 796–804. ISSN: 1548-7091. DOI: 10.1038/nmeth.2016. pmid: 22796662.

- [8] Malte D Luecken and Fabian J Theis. "Current Best Practices in Single-Cell RNA-Seq Analysis: A Tutorial". In: *Molecular Systems Biology* 15.6 (June 1, 2019), e8746. ISSN: 1744-4292. DOI: 10.15252/msb.20188746.
- [9] Catalina A. Vallejos et al. "Normalizing Single-Cell RNA Sequencing Data: Challenges and Opportunities". In: *Nature Methods* 14.6 (6 June 2017), pp. 565–571. ISSN: 1548-7105. DOI: 10.1038/nmeth.4292.
- [10] Daniel T. Gillespie. "Exact Stochastic Simulation of Coupled Chemical Reactions". In: *The Journal of Physical Chemistry* 81.25 (Dec. 1, 1977), pp. 2340–2361. ISSN: 0022-3654. DOI: 10.1021/j100540a008.
- [11] Luke Zappia, Belinda Phipson, and Alicia Oshlack. "Splatter: Simulation of Single-Cell RNA Sequencing Data". In: *Genome Biology* 18 (Sept. 2017), p. 174. ISSN: 1474-760X. DOI: 10.1186/s13059-017-1305-0.
- [12] Beate Vieth et al. "powsimR: Power Analysis for Bulk and Single Cell RNA-Seq Experiments". In: *Bioinformatics* 33.21 (Nov. 1, 2017), pp. 3486–3488. ISSN: 1367-4803. DOI: 10.1093/bioinformatics/btx435.
- [13] Nikolaos Papadopoulos, Parra R. Gonzalo, and Johannes Söding. "PROSSTT: Probabilistic Simulation of Single-Cell RNA-Seq Data for Complex Differentiation Processes". In: *Bioinformatics (Oxford, England)* 35.18 (Sept. 15, 2019), pp. 3517–3519. ISSN: 1367-4811. DOI: 10.1093/bioinformatics/btz078. pmid: 30715210.
- [14] Xiuwei Zhang, Chenling Xu, and Nir Yosef. "Simulating Multiple Faceted Variability in Single Cell RNA Sequencing". In: *Nature Communications* 10.1 (June 13, 2019), pp. 1–16. ISSN: 2041-1723. DOI: 10.1038/s41467-019-10500-w.
- [15] Kelly Street et al. "Slingshot: Cell Lineage and Pseudotime Inference for Single-Cell Transcriptomics". In: *BMC Genomics* 19.1 (June 2018), p. 477. ISSN: 1471-2164. DOI: 10.1186/s12864-018-4772-0.
- [16] R. Gonzalo Parra et al. "Reconstructing Complex Lineage Trees from scRNA-Seq Data Using MERLoT". In: *Nucleic Acids Research* 47.17 (Sept. 26, 2019), pp. 8961–8974. ISSN: 1362-4962. DOI: 10.1093/nar/gkz706. pmid: 31428793.
- [17] Edroaldo Lummertz da Rocha et al. "Reconstruction of Complex Single-Cell Trajectories Using CellRouter". In: *Nature Communications* 9.1 (Mar. 1, 2018), p. 892. ISSN: 2041-1723. DOI: 10.1038/s41467-018-03214-y.
- [18] Yingxin Lin et al. "scClassify: Sample Size Estimation and Multiscale Classification of Cells Using Single and Multiple Reference". In: *Molecular Systems Biology* 16.6 (June 1, 2020), e9389. ISSN: 1744-4292. DOI: 10.15252/msb.20199389.
- [19] Angelo Duò, Mark D. Robinson, and Charlotte Soneson. "A Systematic Performance Evaluation of Clustering Methods for Single-Cell RNA-Seq Data". In: *F1000Research* 7 (2018), p. 1141. ISSN: 2046-1402. DOI: 10.12688/f1000research.15666.2. pmid: 30271584.
- [20] Wouter Saelens et al. "A Comparison of Single-Cell Trajectory Inference Methods". In: *Nature Biotechnology* 37 (May 2019). ISSN: 15461696. DOI: 10.1038/s41587-019-0071-9.
- [21] Charlotte Soneson and Mark D. Robinson. "Bias, Robustness and Scalability in Single-Cell Differential Expression Analysis". In: *Nature Methods* 15.4 (Apr. 2018), pp. 255–261. ISSN: 1548-7105. DOI: 10.1038/nmeth.4612. pmid: 29481549.
- [22] Charlotte Soneson and Mark D. Robinson. "Towards Unified Quality Verification of Synthetic Count Data with countsimQC". In: *Bioinformatics* 34.4 (Feb. 15, 2018), pp. 691–692. ISSN: 1367-4803. DOI: 10.1093/bioinformatics/btx631.
- [23] Vân Anh Huynh-Thu et al. "Inferring Regulatory Networks from Expression Data Using Tree-Based Methods". In: *PLoS ONE* 5.9 (Jan. 2010), e12776. ISSN: 1932-6203. DOI: 10.1371/journal.pone.0012776. pmid: 20927193.
- [24] Thomas Moerman et al. "GRNBoost2 and Arboreto: Efficient and Scalable Inference of Gene Regulatory Networks". In: *Bioinformatics* 35.12 (June 1, 2019), pp. 2159–2161. ISSN: 1367-4803. DOI: 10.1093/bioinformatics/bty916.

REVIEWERS' COMMENTS

Reviewer #2 (Remarks to the Author):

All points were addressed very well, thank you! In particular, the scalability analysis and improved documentation are useful for the community, and very appreciated.

I have two minor points left, which I think the authors can address quickly:

- Results changed for velocity benchmarks – are these robust overall or is it due to the difficulty settings?
- “This means that predicting the RNA velocity for any particular gene is challenging, but the combined information is very informative...”. I agree that the combined information can be informative even if not all genes yield a robust fit. Yet, this is not necessarily because a per-gene fit is challenging, but imho rather due to low intrinsic dimensionality, as only a subset of genes is informative after all. It is also not clear, whether one can make that statement from the correlation metric as, e.g., if correlating two expression profiles, one would also generally obtain rather low correlation scores.

Minor comment 1

Results changed for velocity benchmarks – are these robust overall or is it due to the difficulty settings?

Between the first and second submission, we indeed removed the different difficulty settings from the benchmark because, in hindsight, the ‘easy’ and ‘medium’ difficulty settings had unrealistically high transcription rates. We decided to only use the ‘hard’ difficulty, which uses dyngen’s default parameters. This has the effect of streamlining the benchmark analysis, making it easier for readers to understand the figure and code.

Minor comment 2

“This means that predicting the RNA velocity for any particular gene is challenging, but the combined information is very informative...”. I agree that the combined information can be informative even if not all genes yield a robust fit. Yet, this is not necessarily because a per-gene fit is challenging, but imho rather due to low intrinsic dimensionality, as only a subset of genes is informative after all. It is also not clear, whether one can make that statement from the correlation metric as, e.g., if correlating two expression profiles, one would also generally obtain rather low correlation scores.

This is an interesting and valid remark. I think the challenge herein lies in identifying which subset of genes is informative in deriving a good intrinsic dimensionality. Doing so might result in significantly improved velocity arrow cosine scores.

We added the following text to the manuscript:

Note that, given that some genes are more informative in determining the overall directionality of cell progression, performing a feature selection before computing the embedded dimensionality reduction might result in significantly improved velocity arrow cosine scores.